# Flow-dependent observation errors for GHG inversions in an ensemble Kalman smoother

Michael Steiner[1], Luca Cantarello[2], Stephan Henne[1], and Dominik Brunner[1]

[1]Empa, Swiss Federal Laboratories for Materials Science and Technology, Dübendorf, Switzerland
[2]European Centre for Medium-Range Weather Forecasts, Bonn, Germany

**Correspondence:** Michael Steiner (michael.steiner@empa.ch) and Dominik Brunner (dominik.brunner@empa.ch)

**Abstract.**

Atmospheric inverse modeling is the process of estimating emissions from atmospheric observations by minimizing a cost function, which includes a term describing the difference between simulated and observed concentrations. The minimization of this difference is typically limited by uncertainties in the atmospheric transport model rather than by uncertainties in the observations. In this study, we showcase how a temporally varying, flow-dependent atmospheric transport uncertainty can enhance the accuracy of emission estimation through idealized experiments using the CTDAS-ICON-ART ensemble Kalman smoother system. We use the estimation of European $CH_4$ emissions from the in-situ measurement network as an example, but we also demonstrate the additional benefits for trace gases with more localized sources, such as $SF_6$. The uncertainty in flow-dependent transport is determined using meteorological ensemble simulations that are perturbed by physics and driven at the boundaries by an analysis ensemble from a global meteorology and $CH_4$ simulation. The impact of a direct representation of temporally varying transport uncertainties in atmospheric inversions is then investigated in an observation system simulation experiment framework in various setups and for different flux signals. We show that the uncertainty in the transport model varies significantly in space and time, and it is generally highest during nighttime. We apply inversions using only afternoon observations as is common practice, but also explore the option of assimilating hourly data irrespective of the hour of day using a filter based on transport uncertainty and taking into account the temporal covariances. Our findings indicate that incorporating flow-dependent uncertainties in inversion techniques leads to more accurate estimates of GHG emissions. Differences between estimated and true emissions could be reduced by 9% to 82% more effectively, with generally larger improvements for the $SF_6$ inversion problem and for the more challenging setup with small flux signals.

## 1 Introduction

Atmospheric greenhouse gas (GHG) inversions use observed atmospheric GHG concentrations to estimate surface fluxes. This helps to verify emission reduction targets or the fulfillment of the Paris Agreement, monitor substances whose emissions are prohibited or regulated by the Montreal Protocol, and understand global and regional carbon cycles. The independence of such top-down inversions could also support the development of future emission inventories. However, GHG inversions are subject to various uncertainties, which are expressed in discrepancies among inversion results (e.g., Brunner et al., 2017; Monteil et al.,

2020; Petrescu et al., 2023). Current inversion systems are mostly based on Bayes' theorem and solve the inversion problem by minimizing a cost function with two components: one component penalizes deviations from the a priori state, while the other penalizes differences between simulated and observed concentrations. The weighting of the two terms is determined by corresponding error covariance matrices, which define the magnitude and correlation structure of errors related to these two components. Errors in the second term, commonly denoted as **R** and referred to as observation representation error or model-

data mismatch error (mdm), include all processes that contribute to discrepancies between observations and model, such as aggregation and representation errors, measurement errors, and atmospheric transport errors (Kaminski et al., 2001; Engelen et al., 2002). A major source of uncertainty arises from the transport uncertainty of the atmospheric transport model (Munassar et al., 2023). Errors in the atmospheric transport lead to corresponding errors in modeled tracer concentrations, which may ultimately lead to erroneous flux estimates if not properly accounted for. Constructing the mdm is therefore a crucial yet

challenging task, especially given the limited understanding of the errors associated with the representation of atmospheric transport in numerical models. Although these errors are recognized as significant, there still remains a gap in adequately addressing them within inversion systems, mainly due to the fact that determining atmospheric transport uncertainties typically requires expensive meteorological ensemble simulations.

The importance of transport errors in inverse emission estimation was recognized already by Enting (1993) but system-

atic studies of their impacts and possible solutions were published only much later. Lin and Gerbig (2005) tested the effects of unaccounted wind errors, directly determined from radiosonde data, on inversion results, and demonstrated that these uncertainties can result in significantly biased flux estimates. In a subsequent study, the authors focused on potential biases in inversion results associated with a misrepresentation of vertical mixing in the atmospheric boundary layer (Gerbig et al., 2008). Lauvaux et al. (2009) investigated transport errors in the context of inversions, determining covariances between two stations

in South-West France from a meteorological ensemble simulation, and showed that the effective information content of the observations in an inversion is significantly reduced when considering these covariances. Several subsequent studies investigated transport errors and their characteristics for greenhouse gases, especially $CO_2$, without directly examining the impacts in an inversion framework. Using global simulations, Liu et al. (2011) demonstrated that the $CO_2$ transport uncertainty is highest in the tropics and in regions with the highest emissions from fossil fuels. Miller et al. (2015) examined the importance of transport

error compared to emission signals, analyzing monthly biases and associating them with meteorological conditions. Further studies explored the predictability of $CO_2$ (Polavarapu et al., 2016; Kim et al., 2021) or the significance of errors in transport and boundary conditions for estimating the terrestrial carbon sink in limited area simulations (LAM) (Feng et al., 2019). Two recent studies provided more detailed investigations of transport error and its characteristics. Chen et al. (2019) focused on a sensitivity analysis of transport errors to errors in emission fluxes and initial and boundary data in LAM simulations, while

McNorton et al. (2020) focused on the biogenic feedback to transport error in global simulations. A comprehensive study on the impacts of transport uncertainties on inverse $CO_2$ estimation in an urban context was recently presented by Ghosh et al. (2021). They tested various transport uncertainties in synthetic $CO_2$ flux inversions with pseudo-observations in an urban area with a dense observation network. They compared conventional parameterizations with transport uncertainties sampled from an ensemble of simulations, both with and without spatio-temporal covariances. They demonstrated the importance of con-

sidering covariances in transport uncertainties when using data from a dense network. A method to partially account for the effects of different meteorological situations on the transport uncertainty was applied in Bergamaschi et al. (2022). In their European $CH_4$ inversions they parameterized the mdm as a function of wind speed. Their approach assigns larger uncertainties to observations at low wind speeds, assuming that under such conditions, local emissions may have a greater influence on the observed concentrations but are not well represented in the model.

This paper presents a comprehensive examination of flow-dependent transport uncertainties and proposes a simple method for including them in atmospheric inversions. The method is showcased for the estimation of European $CH_4$ inversions using the ensemble Kalman smoother system CTDAS-ICON-ART introduced by Steiner et al. (2024). Making use of meteorological ensemble simulations, we investigate the spatial, vertical, and temporal characteristics of the transport error for $CH_4$ and investigate the impact of realistic transport uncertainties on inversion results in an observation system simulation experiment

(OSSE). We use synthetically generated $CH_4$ observations ("pseudo-observations") mimicking the observations from the current network of European in-situ stations, but also demonstrate the advantages for trace gases characterized by a more irregular emission distribution, such as $SF_6$. We evaluate the new approach in different setups and for various flux signals. Furthermore, we demonstrate how information on flow-dependent uncertainties may be used to assimilate hourly observations as opposed to afternoon observations only, as frequently done. For this we use a filter based on transport uncertainty and take into account

the temporal covariances. Finally we also demonstrate the impacts of the new transport uncertainty on European $CH_4$ emission estimates using real observations. All inversions presented in this study are listed in Table 1 and 2.

## 2 Model description and methodology

We conducted atmospheric transport simulations of $CH_4$ using the Icosahedral Nonhydrostatic (ICON) atmospheric modeling framework (Wan et al., 2013; Zängl et al., 2015; Pham et al., 2021) with the ART extension for passive and reactive tracers

(Rieger et al., 2015; Weimer et al., 2017; Schröter et al., 2018). Our simulations were configured following the setup described in Steiner et al. (2024), employing an R3B6 limited area grid ($\Delta x \approx 26$ km) covering Europe with 60 vertical levels. In the simulations, we use a time step of 120 seconds and nudge the simulation weakly towards the driving reanalysis data. In the inversion step, the 21344 emission regions (grid cells) in the state vector are optimized using the fixed-lag ensemble Kalman smoother implemented in CTDAS with an assimilation window length of 9 days and a lag of 2. The number of members in the

emissions ensemble is 192. Unlike our previous study with a single forward simulation, we created a meteorological ensemble (see Sect. 2.1) with 10 ensemble members driven by perturbed meteorological boundary conditions and model physics, as well as by perturbed $CH_4$ boundary conditions from the same global ensemble simulation that provided the meteorological boundary conditions. Each ensemble member contains two $CH_4$ tracers: a background tracer ($CH_4^{bg}$) representing the perturbed $CH_4$ boundary conditions, and an emission tracer ($CH_4^{emis}$) representing the additional $CH_4$ emitted within our European model

domain. These $CH_4^{emis}$ tracers experienced the same (unperturbed) $CH_4$ emissions such that the ensemble of 10 $CH_4^{tot} = CH_4^{bg} + CH_4^{emis}$ tracers solely represents the effect of transport uncertainty.

## 2.1 ICON-ART ensemble simulations

To generate a meteorological ensemble, we ran the model with 10 members, each driven by the output of an experimental Ensemble of Data Assimilation (EDA) simulation conducted at ECMWF (experiment ID "hyfd"), which included global surface emissions inversion and transport of $CO_2$ and $CH_4$ (McNorton et al., 2020). The experiment was conducted for the month of July 2019 and was carried out using the IFS cycle 48r1 with 10 ensemble members at 25 km resolution (Tco399), in which model physics (in the form of Stochastically Perturbed Parameterization Tendencies (SPPT, Leutbecher et al., 2017)), observations (both, weather and GHG observations), and sea surface temperature were perturbed. Additionally, the GHG emissions were also perturbed, with the perturbations sampled from a log-normal distribution with an a priori uncertainty of 100 %. The perturbations were not spatially correlated but a correlation length of 100 km was assumed in the background covariance matrix for $CH_4$ emissions. The background error covariance matrix for GHG concentrations was static, that is, it was not updated during the experiment, and was based on differences between forecasts with different lead times obtained from a climatological sample (cf. NMC method Parrish and Derber, 1992). The background error covariance matrix for NWP fields was based on the archived IFS EDA o-suite (experiment 0001) for the same period. Greenhouse Gases Observing Satellite (GOSAT), Infrared Atmospheric Sounding Interferometer (IASI), and TROPOspheric Monitoring Instrument (TROPOMI) retrievals of $CH_4$ were assimilated. The emissions were optimized independently in each 12-hour window.

In the ICON ensemble simulations, in addition to the perturbed driving data (meteorological variables + $CH_4$ concentrations to drive the $CH_4^{BG}$ tracer), we also applied perturbations to model physics tuning parameters as implemented in ICON for the ensemble data assimilation scheme of the German weather service. Together, these perturbations are expected to represent the typical level of uncertainties present in state-of-the-art meteorological analysis products.

The perturbed $CH_4$ concentration fields of the driving data was used to initialize and drive the background tracers $CH_4^{bg}$ in our simulations. In this study, we did not optimize the background concentrations, but the perturbations of the different background fields were part of the artificial transport error. In addition to the $CH_4^{bg}$ tracer, we also incorporated a $CH_4^{emis}$ tracer into our ensemble simulations using ICON-ART, transporting the emitted signal with (unperturbed) emissions introduced via the online emission module (OEM) (Jähn et al., 2020; Steiner et al., 2024). The computational cost, measured in node-hours, for the 10-member ensemble simulation was about 1.7 times the cost of regular inversions. This means that the total computational cost (a priori meteorological ensemble and inversion) was 2.7 times the cost of a regular inversion, which is a considerable but not prohibitive increase.

## 2.2 CTDAS inversion setup

The setup of the CTDAS-ICON-ART inversion system aligns with Steiner et al. (2024). We optimize the emission scaling factors in each grid cell of the R3B6 grid while applying exponential decaying correlations with a length scale of 200 km in the a priori error covariance matrix. However, optimization is restricted to a single emission category, namely the total $CH_4$ emissions, which is the sum of anthropogenic and various natural emissions.

Since we compare different approaches in representing the mdm uncertainty, the question arises, how these uncertainties should be scaled to enable a fair comparison. Since the innovation chi-square statistics is a common diagnostic to judge the validity of the uncertainty assumptions made in an inversion (Berchet et al., 2015; Michalak et al., 2017), we contend that comparability is best achieved when the uncertainties in each inversion are scaled such that the innovation chi-square value is 1. The chi-square metric delineates the ratio of a priori residuals between observed and simulated concentrations to the total a priori variance in the observation space, accounting for the projected a priori variance to the observation space alongside the model-data mismatch

$$\chi^2_{innov} = \frac{1}{n} \sum (\mathbf{y^o} - \mathbf{H}(\mathbf{x^b}))^T (\mathbf{H}\mathbf{P^b}\mathbf{H} + \mathbf{R})^{-1} (\mathbf{y^o} - \mathbf{H}(\mathbf{x^b})) \tag{1}$$

Here, $\mathbf{y^o}$ denotes observed data, $\mathbf{H}$ the observation operator, $\mathbf{x^b}$ the a priori (background) state, $\mathbf{P^b}$ the a priori error covariance matrix, $\mathbf{R}$ the observation error covariance matrix (or mdm), and $n$ the number of observations. The requirement of a chi-square value close to 1 results in some differences in the magnitude of the mdm (indicated as $\alpha$ in Table 1 and 2), but we believe this is a better approach than requiring, for example, that the mean mdm is identical between different inversions.

## 2.3 Idealized Experiments

In our idealized setup, one of these 10 ensemble members was considered to represent the "true" meteorology and used to generate the pseudo-observations of $CH_4$. All inversions were then performed using a different ensemble member in order to mimic the fact that the simulated meteorology (and the corresponding transport of $CH_4$) in general deviates from the true meteorology within the range of uncertainty of state-of-the-art meteorological analyses.

The ensemble spread (corresponding to the standard deviation) of the $CH_4^{tot}$ tracers was sampled at each station location and used to determine a temporally varying, flow-dependent model-data mismatch (mdm) replacing the static mdm used in our previous inversion study (Steiner et al., 2024). In contrast to our previous study, we optimized only the emissions in the idealized setup. We did not optimize the background concentrations because the differences in background $CH_4$ concentrations introduced by deviating from "true" meteorology and using perturbed background $CH_4^{bg}$ concentrations in the driving data are part of the artificially created transport error and contribute to the ensemble spread that determines the mdm. However, if there were systematic biases in background $CH_4$ in an application with real data, it would still be necessary to optimize background concentrations together with the emissions. The performance of the inversions with the flow-dependent mdm was compared with inversions using the static mdm as used in Steiner et al. (2024). To ensure a fair comparison, the mdm was scaled in each inversion so that the innovation chi-squared value in each inversion was 1.

In the reference setup (inversions "fc01" and "fc02"), we utilized an a priori variance of 0.07 (unitless) at grid cell level, which corresponds to an emission uncertainty of 26%. With such a low emission signal it is easier to demonstrate the benefits of our new approach as the relative importance of transport uncertainties is larger. Since the same variance of 0.07 was used for generating the pseudo-observations (see Sect. 2.5), the reference setup corresponds to the optimal situation where the assumed a priori uncertainties are in perfect agreement with the true emission errors.

**Table 1.** Overview of all inversions used in this study. The column "mdm" indicates whether the flow-dependent mdm ("FD") or the constant mdm ("C") is used. $\alpha$ is the scaling factor from Eq. 2. The column "emis" indicates whether the $CH_4$ emissions or $SF_6$ emissions are used. "ER" stands for error reduction.

| ID | mdm | emis | $\sigma_{true}$ | $\sigma_{prio}$ | $\alpha$ | ER $\frac{kg}{s}$ | ER % | ID | mdm | emis | $\sigma_{true}$ | $\sigma_{prio}$ | $\alpha$ | ER $\frac{kg}{s}$ | ER % |
|---|---|---|---|---|---|---|---|---|---|---|---|---|---|---|---|
| fc01 | FD | $CH_4$ | 0.07 | 0.07 | 1.28 | 36.57 | 15.8 | fs01 | FD | $SF_6$ | 0.07 | 0.07 | 1.66 | 55.56 | 24.9 |
| fc02 | C | $CH_4$ | 0.07 | 0.07 | 1.44 | 20.12 | 8.7 | fs02 | C | $SF_6$ | 0.07 | 0.07 | 2.08 | 30.77 | 13.8 |
| fc03 | FD | $CH_4$ | 0.25 | 0.25 | 1.91 | 88.69 | 20.3 | fs03 | FD | $SF_6$ | 0.25 | 0.25 | 2.52 | 123.69 | 29.3 |
| fc04 | C | $CH_4$ | 0.25 | 0.25 | 1.96 | 66.29 | 15.2 | fs04 | C | $SF_6$ | 0.25 | 0.25 | 2.55 | 91.69 | 21.7 |
| fc05 | FD | $CH_4$ | 0.56 | 0.56 | 3.04 | 142.29 | 21.8 | fs05 | FD | $SF_6$ | 0.56 | 0.56 | 3.63 | 196.20 | 31.0 |
| fc06 | C | $CH_4$ | 0.56 | 0.56 | 2.54 | 124.36 | 19.0 | fs06 | C | $SF_6$ | 0.56 | 0.56 | 3.25 | 159.54 | 25.2 |
| fc07 | FD | $CH_4$ | 0.71 | 0.71 | 3.46 | 160.97 | 21.9 | fs07 | FD | $SF_6$ | 0.71 | 0.71 | 3.99 | 220.78 | 31.0 |
| fc08 | C | $CH_4$ | 0.71 | 0.71 | 2.81 | 144.57 | 19.6 | fs08 | C | $SF_6$ | 0.71 | 0.71 | 3.52 | 186.86 | 26.3 |
| fc09 | FD | $CH_4$ | 0.87 | 0.87 | 1.90 | 181.19 | 22.2 | fs09 | FD | $SF_6$ | 0.87 | 0.87 | 4.30 | 251.67 | 31.9 |
| fc10 | C | $CH_4$ | 0.87 | 0.87 | 2.10 | 165.41 | 20.3 | fs10 | C | $SF_6$ | 0.87 | 0.87 | 3.70 | 211.05 | 26.8 |
| fc11 | FD | $CH_4$ | 1.00 | 1.00 | 3.95 | 194.12 | 22.2 | fs11 | FD | $SF_6$ | 1.00 | 1.00 | 4.60 | 267.32 | 31.7 |
| fc12 | C | $CH_4$ | 1.00 | 1.00 | 3.24 | 178.22 | 20.4 | fs12 | C | $SF_6$ | 1.00 | 1.00 | 3.99 | 229.92 | 27.2 |
| uc01 | FD | $CH_4$ | 0.07 | 0.70 | 1.14 | -13.80 | -6.0 | us01 | FD | $SF_6$ | 0.07 | 0.70 | 1.53 | -9.19 | -4.1 |
| uc02 | C | $CH_4$ | 0.07 | 0.70 | 1.28 | -49.98 | -21.6 | us02 | C | $SF_6$ | 0.07 | 0.70 | 1.854 | -116.26 | -52.0 |
| uc03 | FD | $CH_4$ | 0.07 | 0.40 | 1.22 | 13.94 | 6.0 | us03 | FD | $SF_6$ | 0.07 | 0.40 | 1.60 | 28.92 | 12.9 |
| uc04 | C | $CH_4$ | 0.07 | 0.40 | 1.27 | -21.65 | -9.4 | us04 | C | $SF_6$ | 0.07 | 0.40 | 1.93 | -43.58 | -19.5 |
| uc05 | FD | $CH_4$ | 0.07 | 0.12 | 1.20 | 34.43 | 14.9 | us05 | FD | $SF_6$ | 0.07 | 0.12 | 1.54 | 52.94 | 23.7 |
| uc06 | C | $CH_4$ | 0.07 | 0.12 | 1.40 | 13.71 | 5.9 | us06 | C | $SF_6$ | 0.07 | 0.12 | 2.02 | 22.45 | 10.0 |
| uc07 | FD | $CH_4$ | 0.07 | 0.07 | 1.28 | 36.57 | 15.8 | us07 | FD | $SF_6$ | 0.07 | 0.07 | 1.66 | 55.56 | 24.9 |
| uc08 | C | $CH_4$ | 0.07 | 0.07 | 1.44 | 20.12 | 8.7 | us08 | C | $SF_6$ | 0.07 | 0.07 | 2.08 | 30.77 | 13.9 |
| uc09 | FD | $CH_4$ | 0.07 | 0.05 | 1.26 | 36.38 | 15.7 | us09 | FD | $SF_6$ | 0.07 | 0.05 | 1.60 | 55.24 | 24.7 |
| uc10 | C | $CH_4$ | 0.07 | 0.05 | 1.41 | 22.31 | 9.6 | us10 | C | $SF_6$ | 0.07 | 0.05 | 2.73 | 31.71 | 14.2 |
| uc11 | FD | $CH_4$ | 0.07 | 0.037 | 1.20 | 35.75 | 15.5 | us11 | FD | $SF_6$ | 0.07 | 0.035 | 1.64 | 53.91 | 24.1 |
| uc12 | C | $CH_4$ | 0.07 | 0.037 | 1.40 | 23.86 | 10.3 | us12 | C | $SF_6$ | 0.07 | 0.035 | 2.13 | 31.11 | 13.9 |
| uc13 | FD | $CH_4$ | 0.07 | 0.0175 | 1.26 | 32.85 | 14.2 | us13 | FD | $SF_6$ | 0.07 | 0.0175 | 1.66 | 48.88 | 21.9 |
| uc14 | C | $CH_4$ | 0.07 | 0.0175 | 1.44 | 23.39 | 10.1 | us14 | C | $SF_6$ | 0.07 | 0.0175 | 2.20 | 26.84 | 120 |
| uc15 | FD | $CH_4$ | 0.07 | 0.007 | 1.28 | 27.27 | 11.8 | us15 | FD | $SF_6$ | 0.07 | 0.007 | 1.78 | 39.52 | 17.7 |
| uc16 | C | $CH_4$ | 0.07 | 0.007 | 1.41 | 21.08 | 9.1 | us16 | C | $SF_6$ | 0.07 | 0.007 | 2.24 | 20.10 | 9.0 |

Starting from the reference setup, two sets of sensitivity experiments were conducted. In the first set, different combinations of a priori variances ranging from 0.007 to 0.7 were tested in order to analyze the impact of a priori assumptions deviating

**Table 2.** Continuation of Table 1 for inversions with the synthetic setup and hourly observations as well as for inversions with real observations.

| ID | mdm | emis | $\sigma_{true}$ | $\sigma_{prio}$ | $\alpha$ | ER $\frac{\text{kg}}{\text{s}}$ | ER % | remark |
|---|---|---|---|---|---|---|---|---|
| $R_c$ | C | $CH_4$ | 0.07 | 0.07 | 0.73 | 36.63 | 15.8 | hourly obs., diag. R |
| $R_d$ | FD | $CH_4$ | 0.07 | 0.07 | 0.80 | 40.89 | 17.7 | hourly obs., diag. R |
| $R_e$ | FD | $CH_4$ | 0.07 | 0.07 | 0.80 | 45.90 | 19.9 | hourly obs., incl. covariances |
| real_c | C | $CH_4$ | - | 0.6 | station-dependent | - | - | real observations |
| real_f | FD | $CH_4$ | - | 0.6 | station-dependent | - | - | real observations |

from the true emission error (inversions "uc01" to "uc16" and "us01" to "us16"). In the second set, the true variances were varied between 0.07 and 1.0 in order to assess the potential of improving the inversion results using our proposed method for different ratios between flux uncertainties and transport uncertainties. All inversions were performed for two different emission fields. One emission field (in the inversions "fc01" to "fc16" and "uc01" to "uc16") is the same as in Steiner et al. (2024), which comprises the sum of anthropogenic emissions from EDGARv6 (Crippa et al., 2021) and various natural sources: peatlands and mineral soils from JSBACH-HIMMELI (Jena Scheme for Biosphere-Atmosphere Coupling in Hamburg coupled to HelsinkI Model for Methane build-up and emission for peatlands; Raivonen et al., 2017; Reick et al., 2013) (version 2), inland water provided by Université Libre de Bruxelles to the Global Carbon Project (GCP) $CH_4$ data set; (Saunois et al., 2020), termites (Saunois et al., 2020), ocean (Weber et al., 2019), and biofuels and biomass burning from the Global Fire Emission Database 4.1s (GFED; van der Werf et al., 2017) as well as geological emissions (Etiope et al., 2019) (scaled to a global total of 15 Tg). The second emission field (in the inversions "fs01" to "fs16" and "us01" to "us16") contains the same total amount but is spatially distributed according to the $SF_6$ emission field of the categories "NFE" and "PRU" in EDGARv7 (Crippa et al., 2022). The latter shows a more irregular emission distribution than $CH_4$ and, hence, serves as an additional test case with expectedly larger gradients. Figure 1 shows the spatial distribution of the two emission fields in our model domain.

In another experiment (inversions "fc01" to "fc12" and "fs01" to "fs12"), we compared the aforementioned inversions, where we assimilated only afternoon averages (nighttime averages for mountain stations), with inversions where we assimilated hourly observations filtered based on the ensemble spread of $CH_4$ concentrations. The filter excluded observations during times when the ensemble spread exceeded 5 ppb. This resulted in the exclusion of ca. 22% of the 19.530 observations. Figure 2 shows the fraction of excluded observations as a function of hour of day. For these inversions, however, the assumption of temporally uncorrelated errors was no longer valid and were accounted for by introducing off-diagonal elements in the mdm error covariance matrix. The temporal error correlations were computed from the ensemble spread. For each of the hourly observations, the error correlation with the observations at the same station in the next 36 hours were considered.

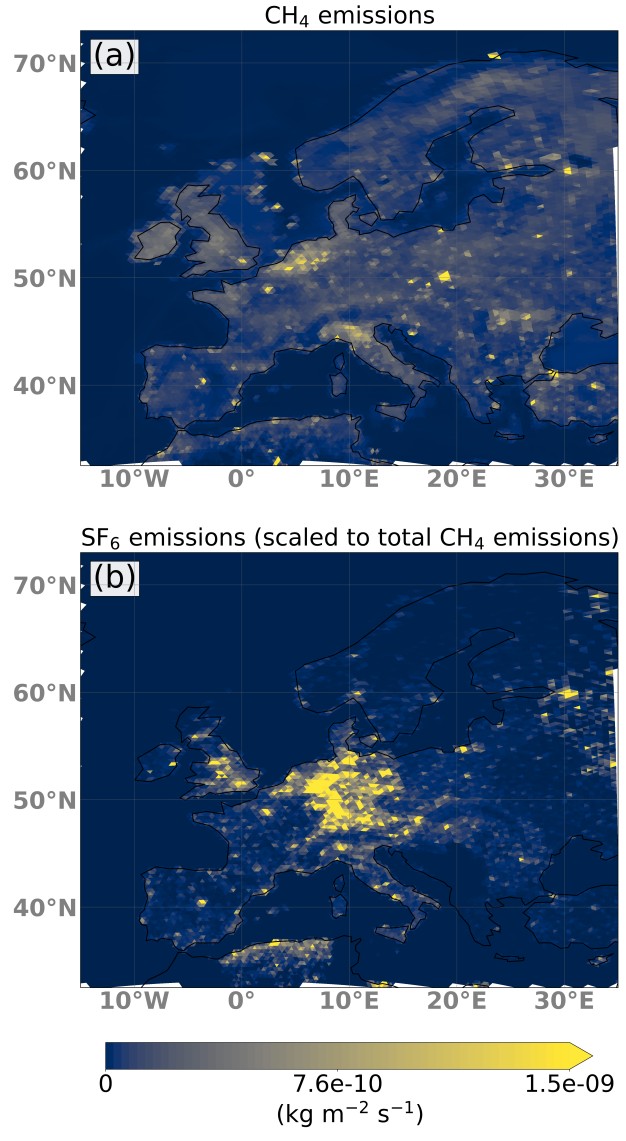

**Figure 1.** Spatial distribution of $CH_4$ emissions (a) as well as $CH_4$ re-distriubted to follow EDGAR $SF_6$ emissions (b) remapped onto the simulation grid. The emissions are representative for the period of 02–11 July 2019.

## 2.4 Experiment with real-data

The setup for the application with real data (inversions "real_c" and "real_f") closely followed the setup for the idealized experiment with the assimilation of daily afternoon or nighttime means (e.g. as in "fc01"). We used the same setup for the ICON simulations with a grid over Europe and the same state vector in the inversions. The transport uncertainties for the mdm

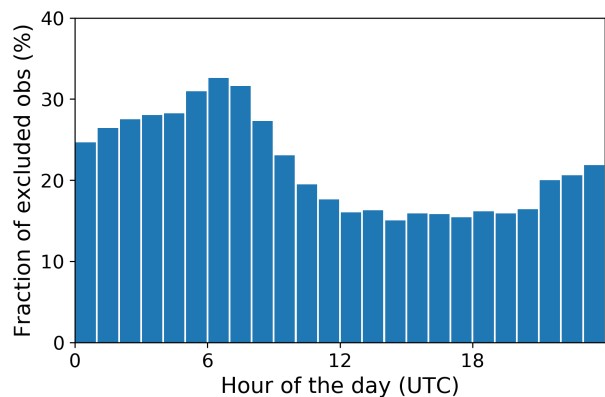

**Figure 2.** The fraction of pseudo-observations that were excluded from all observations available as a function of hour of the day (UTC) by applying a threshold of 5 ppb for the ensemble spread in the ensemble of $CH_4$ concentrations.

were also derived from the same ensemble simulation. However, in order to stay closer to real meteorology and background $CH_4$ concentrations, the forward simulations of the inversion were driven at the domain boundaries by ERA5 reanalysis data for meteorology (Hersbach et al., 2020) and by the CAMSv22r2 product (available via https://ads.atmosphere.copernicus.eu/, last access: 18 April 2024) for background $CH_4$ concentrations. Using the CAMSv22r2 product was necessary as the $CH_4$ mole fractions from the experimental ensemble simulation had too large biases.

### 2.5 Pseudo-observations

Pseudo-observations, following the methodology outlined in Steiner et al. (2024), are generated with a forward simulation of ICON-ART, wherein the $CH_4^{tot}$ tracer concentrations are sampled at the station locations. In this simulation, the emission field is perturbed using a set of "true" scaling factors, which we aim to reproduce as accurately as possible through the inversions starting from unscaled emissions. The "true" scaling factors are a field of spatially correlated random perturbations with a correlation length of 200 km. In our standard configuration, a variance of 0.07 is applied to generate this perturbation field. We systematically explored different configurations, varying the true and a priori variances, with true variances ranging from 0.007 to 1.0. In addition, to mimic measurement noise, a 2 ppb noise was introduced. Pseudo-observations were generated at the stations available in the dataset of the European Obspack 2022-1 L2 release (ICOS RI et al., 2022) for the year 2019.

### 2.6 Real observations

In the application with real observations, the Obspack dataset of quasi-continuous in-situ observations from 28 stations was used. Most of the stations are members of the atmosphere network of the Integrated Carbon Observation System (ICOS) (Heiskanen et al., 2022). As in our previous study, we distinguish mountain sites from sites in flat terrain. Stations where the model topography was more than 200 m lower than the actual topography (due to coarse grid representation) were classified as

mountain stations. For sites in flat terrain, only daytime (11:00 to 16:00 local time) mean values were assimilated, as usually done in atmospheric inverse modeling to avoid difficulties in representing shallow nocturnal boundary layers. In contrast, only nighttime mean values between 23:00 and 06:00 local time were assimilated for mountain sites, as these are least influenced by pollution from daytime up-slope valley winds, which are difficult to represent in a coarse resolution model. The height at which the model output was sampled was different for mountain sites than for sites in flat terrain to account for the fact that the smooth model topography typically underestimates the real altitude of mountain sites. For sites in flat terrain, the (relative) height of the observation above ground was preserved, whereas for mountain sites a height in between the relative height to the model topography and the absolute height of the station was chosen. Mountain stations are indicated in Fig. 8 with triangles, lowland stations with circles. For stations located on a hill but still within the daytime boundary layer (e.g., Beromünster), only measurements in the afternoon were used as for stations in flat terrain, but the vertical sampling of model fields was done in the same way as for mountain stations in order to maintain a realistic relative distance from near-surface emissions.

## 2.7 Observation error

For each pseudo-observation, we calculate the ensemble spread of $\mathrm{CH_4^{tot}}$ in the ensemble. This ensemble spread is then incorporated into the flow-dependent model-data mismatch (mdm) and scaled to achieve an optimal innovation chi-squared value (see Sect. 2.2). Specifically, the flow-dependent mdm is computed as

$$mdm = \alpha \cdot std(\mathrm{CH_4^{tot}}) + 2 \; (ppb) \tag{2}$$

where the factor $\alpha$ varies depending on the inversion (to keep an innovation chi-squared value of 1) but remains constant across all stations and time steps within one inversion. The additional term of 2 ppb accounts for the 2 ppb noise introduced to the pseudo-observations (see Sect.2.5).

Inversions with the flow-dependent mdm are compared with inversions with the static mdm, which was implemented following a similar principle as outlined in Steiner et al. (2024). The static mdm varies between stations but remains constant over time. In this study, we calculated at each station the average of the flow-dependent mdm's for the observations assimilated, adjusting them with a factor to maintain an innovation chi-squared value of 1.

The mdm's for application with real observations was created in the same manner, with the only difference being that the factor **x** was chosen to be station-dependent, which allows to achieve a chi-squared value of 1 for each station separately. This became necessary because, unlike in the synthetic setup, some regions exhibited significantly larger biases in the background concentrations than other regions. With this adjustment, stations where a large bias occurred had a lower weight than stations with good a priori agreement.

For constructing the R-matrix for inversions assimilating hourly observations, we calculated for each station and hour of the day the mean temporal correlation with observations for the next 36 hours during the inversion period. We then fitted a function to these data for each station and hour of the day. The function is a combination of exponential decay and a Gaussian distribution:

$$exp\frac{-\Delta t}{a} + b \, exp\frac{-(\Delta t + c)^2}{2 \, d^2} \tag{3}$$

where the parameters $a$, $b$, $c$ and $d$ are fitted. We chose this fit because it is able to represent the decay in the first hours and the correlated errors between two nights at lowland stations (see Fig. 7). The mean value of the decay time $a$ over all stations and hours is 7.8 hours, with a large variability indicated by a standard deviation of 4.0 hours. The inversion of the R-matrix, which only considers the fitted covariances with the next 36 hours of observations, is very unstable due to poor conditioning. As a result, unrealistic results are produced when inverting $\mathbf{HP^bH + R}$. To address this issue, we conditioned the R-matrix by multiplying the fitted covariances with a factor that exponentially decreases with time ($\exp(-\Delta t/24\text{h})$) as proposed also by Ghosh et al. (2021).

## 3  Results and discussion

### 3.1  Characteristics of the transport error

We start the analysis with illustrating examples of the ratio between flux and transport uncertainties, which can be interpreted as the signal-to-error ratio of the flux signal: Fluxes can only be retrieved reliably if this ratio is larger than one. The ratio was computed as the ratio between the spread in $CH_4^{emis}$ concentrations of the flux ensemble (used in the inversions) to the spread in $CH_4^{tot}$ concentrations of the meteorological ensemble. The time series of this ratio for Cabauw (lowland site) and Monte Cimone (mountain site) at two different magnitudes of flux signals, once with a variance of 0.07 and once with a variance of 1.0, are depicted in Fig. 3. This illustration shows that even in the scenario with a small flux error variance of 0.07, the signal of flux uncertainties is often still stronger than the signal of the transport error (ratio above 1). The low values at the beginning of the time series are due to a spin-up effect: While the emission signal is still extremely small ($CH_4^{emis}$ has not yet reached the stations), the spread in $CH_4^{bg}$ is already fully developed due to the perturbed IC/BC.

A snapshot of the ensemble spread of $CH_4$ at the lowest model levels at an arbitrary time step (Fig. 4) highlights the spatial variability of the transport error. The spread in the tracer of emitted $CH_4^{emis}$ reveals hotspots of large uncertainties, particularly downwind of strong $CH_4$ sources, but also elongated features of high uncertainty likely associated with frontal zones. This underscores the influence that atmospheric flow conditions have on the structure of the transport error. The spread in the background tracer $CH_4^{bg}$ is of similar magnitude but is much smoother.

Further insight into the structure of the transport uncertainty is obtained by plotting time series of vertical profiles of the ensemble spread. Figure 5 shows such a time series for the Dutch station Cabauw. It shows distinct periodic increases in the ensemble spread near the surface during the nights. These increases reflect the uncertainties associated with the simulation of boundary layer processes and their impact on boundary layer heights, which has a particularly strong impact on tracer concentrations in shallow nocturnal boundary layers. The periodic increases in uncertainty are occasionally superimposed by larger-scale events, such as the one from July 9-12. The higher uncertainties at night support the common practice in GHG inversions to use only daytime observations such as afternoon averages. At the same time, it's conceivable that the use of a meteorological ensemble also provides an opportunity to filter observations based on transport uncertainty rather than time of day and thereby to use the information provided by the observations more efficiently.

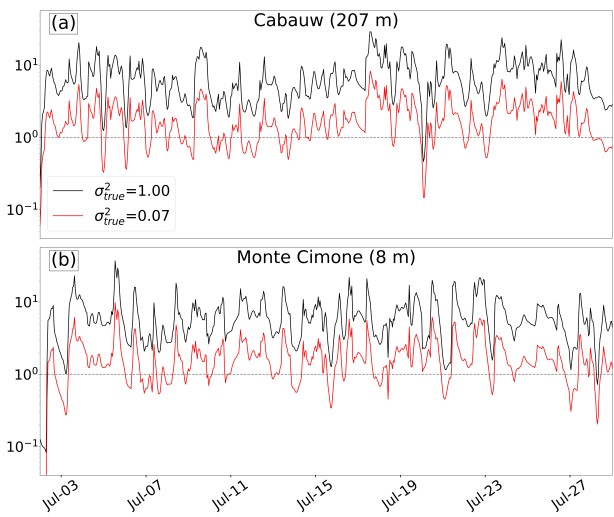

**Figure 3.** Time series of the ratio between emission signal and transport uncertainty at the stations Cabauw (a) and Monte Cimone (b) for two different levels of emission signal.

The mean diurnal ensemble spread of $CH_4$ concentrations is depicted in Fig. 6, distinguishing between lowland and mountain stations. Each figure presents two scenarios: one where the ensemble is generated with perturbed initial and boundary conditions only (IC/BC, gray), and one where in addition to IC/BC also the model physics is perturbed (IC/BC + STTP, red). The spread attributed solely to perturbed IC/BC accounts for approximately 50% of the spread in the two tracers when both perturbed model physics and IC/BC are considered. The higher spread in the nocturnal boundary layer at lowland sites and the peak during the early morning hours is caused approximately equally by the perturbed model physics and by the perturbed IC/BC conditions. At mountain stations, the spread is nearly constant over the day. This indicates that the ensemble is not fully capable of estimating the uncertainty at mountain sites, since a fundamental problem, the misrepresentation of thermally induced flow in the afternoon, is inherent to all ensemble members. This supports the common practice of assimilating only nighttime observations at mountain stations.

The temporal correlations, illustrated in Fig. 7 (in analogy to Fig. 4 in Lauvaux et al., 2009), provide information on the temporal structure of the transport error. Two different periods of the day are examined separately (0 a.m. to 8 a.m., 12 p.m. to 8 p.m) to emphasize the differences between day and nighttime conditions. Each line shows the error correlation (y-axis) of an hourly mean observation with the observations of the next 36 hours (x-axis), with separate analyzes conducted for the lowland and mountain stations. At lowland stations, the nocturnal values exhibit significant error correlations with subsequent night-time observations but lack correlation with daytime values. Similarly to Lauvaux et al. (2009), the correlations with observations from the following night show that, on the one hand, the nocturnal error structures are determined by static parameters that cause similar errors in different nights, but, on the other hand, the system is also sensitive to disturbances, as the correlations remain below 0.5. It can also be seen that the decline in correlations for the 8 a.m. observation occurs earlier

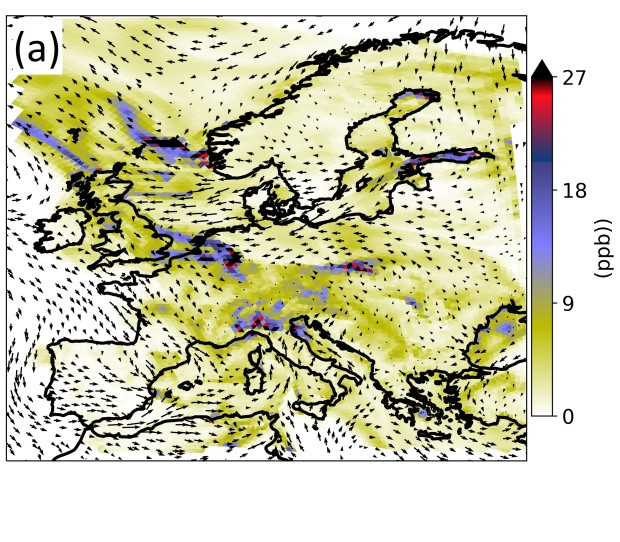

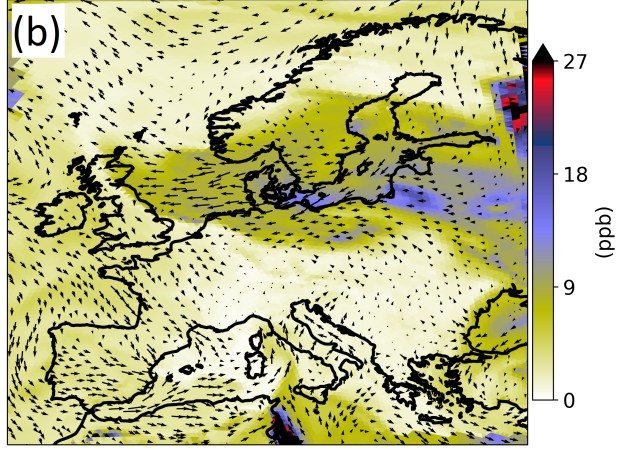

**Figure 4.** Map of the ensemble spread of $CH_4^{emis}$ (a) and $CH_4^{bg}$ (b) in the lowest model level at 2019-07-27 15:00 UTC. The arrows show the ensemble mean wind in the lowest model level.

than for the midnight observation, which is a result of the earlier breakdown of the nocturnal boundary layer for the 8 a.m. observation. For mountain sites, this pattern of recurring error correlations is much less pronounced. The daytime observations at both stations exhibit exponentially decaying correlations without a subsequent increase in the afternoon of the following 290 day, contrasting with the observed nocturnal correlations.

We finally examined the correlations between the $CH_4$ transport uncertainty in our ensemble simulation and the ensemble spread of wind speed and direction, which could also be obtained from a re-analysis product without the need for an ensemble forward simulation. However, our $CH_4$ transport errors show only weak correlations with the ensemble spread of wind speed (0.15 on average, ranging from -0.19 to 0.47 at individual stations) and direction (0.10 on average, ranging from 0.14 to 0.42 at

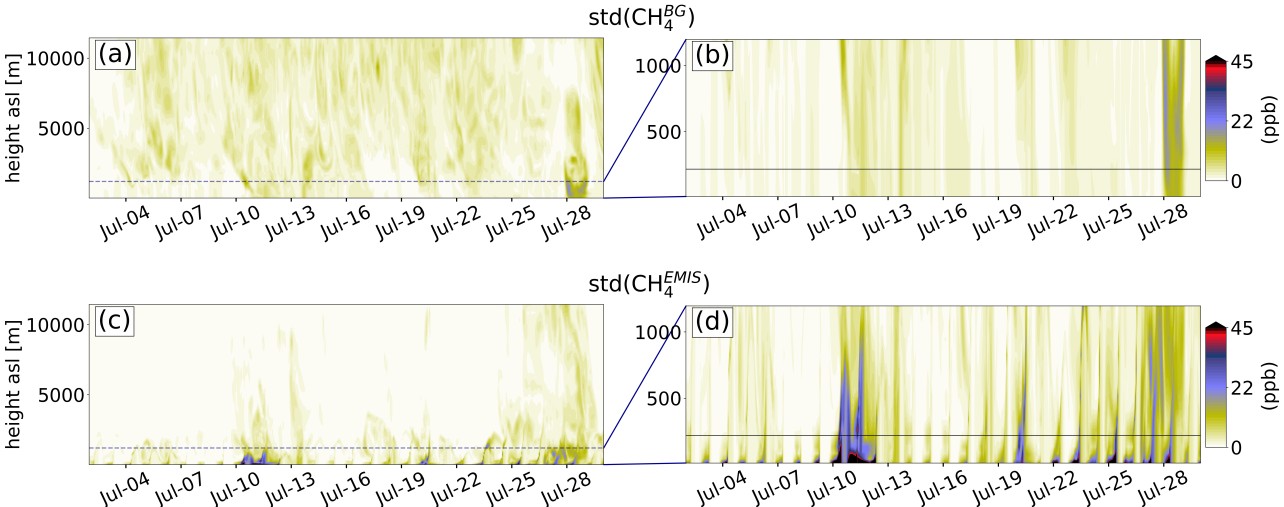

**Figure 5.** Time series of vertical profiles of the standard deviation of $CH_4^{bg}$ (a-b) and $CH_4^{emis}$ (c-d) at Cabauw. The dashed lines in (a and c) indicate the upper boundary of the plots in (b and d). The black line in the right column plots indicates the inlet height at Cabauw.

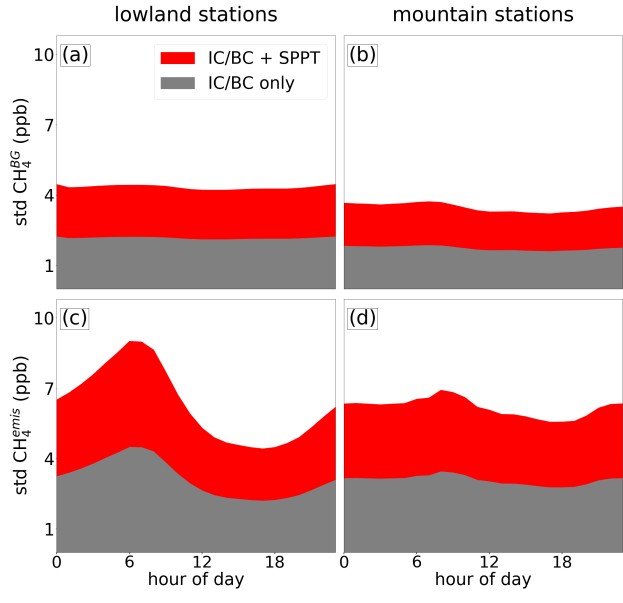

**Figure 6.** Diurnal profiles of the standard deviation of $CH_4^{bg}$ (a and b) and $CH_4^{emis}$ (b and c) concentrations in the ensemble for an ensemble simulation with perturbed IC/BC only (gray) and an ensemble simulation with perturbed IC/BC and perturbed model physics (red).

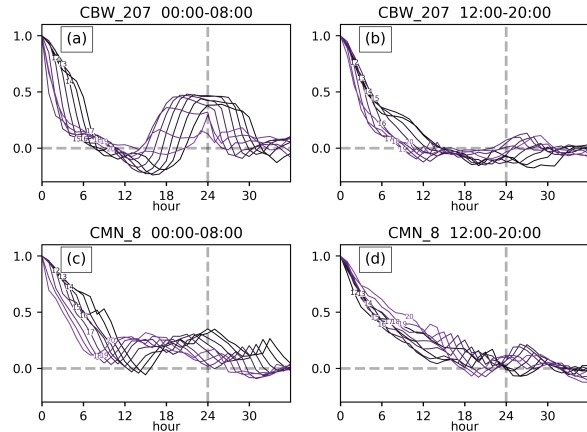

**Figure 7.** Mean temporal correlations at Cabauw (a and b) and Monte Cimone (c and d) for observations at 00 to 08 UTC (a and c) and 12 to 20 UTC (b and d) with observations in the next 36 hours. The error correlations over time (x-axis) for each observation is indicated by one line.

individual stations). This result supports the conclusions of Miller et al. (2015) who discussed and tested the incorporation of transport uncertainty into the mdm without the need for expensive ensemble simulations. They computed correlations between monthly $CO_2$ biases in atmospheric transport (relative to $CO_2$ boundary layer enhancements) and individual meteorological variables in a global ensemble simulation. The strongest correlations were found with inverse temperature over terrestrial regions (0.45) and with zonal winds over the oceans (0.29). However, many errors could not be explained by a single explanatory variable.

### 3.2 Flow-dependent observation error in an idealized setup

Figure 8 shows the true scaling factors alongside the optimized factors obtained from inversions using both the the static mdm ("fc02") and the flow-dependent mdm ("fc01"). Both inversions seem to be similarly successful in reproducing the large-scale patterns of the true state, especially in central Europe where emission fluxes and observation density are high. To better compare the quality of the results, Fig. 9 illustrates the improvement achieved by the flow-dependent mdm compared to the static mdm, both in terms of scaling factors (a) and emission fluxes (b). The predominance of green colors suggests a significant overall improvement with the implementation of the new mdm. Since we perform these inversions in a synthetic setup where the ground truth is known, we can compute the flux error reduction precisely. Summed over the entire domain, this reduction amounts to 20.12 $\mathrm{kg\,s^{-1}}$ (or 8.7% of the a priori total error) with the static mdm and 36.57 $\mathrm{kg\,s^{-1}}$ (or 15.8% of the a priori total error) with the flow-dependent mdm, which corresponds to an improvement by 82%.

Figure 10 illustrates the relationship between error reduction and the ratio of the "true" to the a priori variance for both, the inversions with a flow-dependent and static mdm. In this plot, the true variance remains constant at 0.07 while the a priori

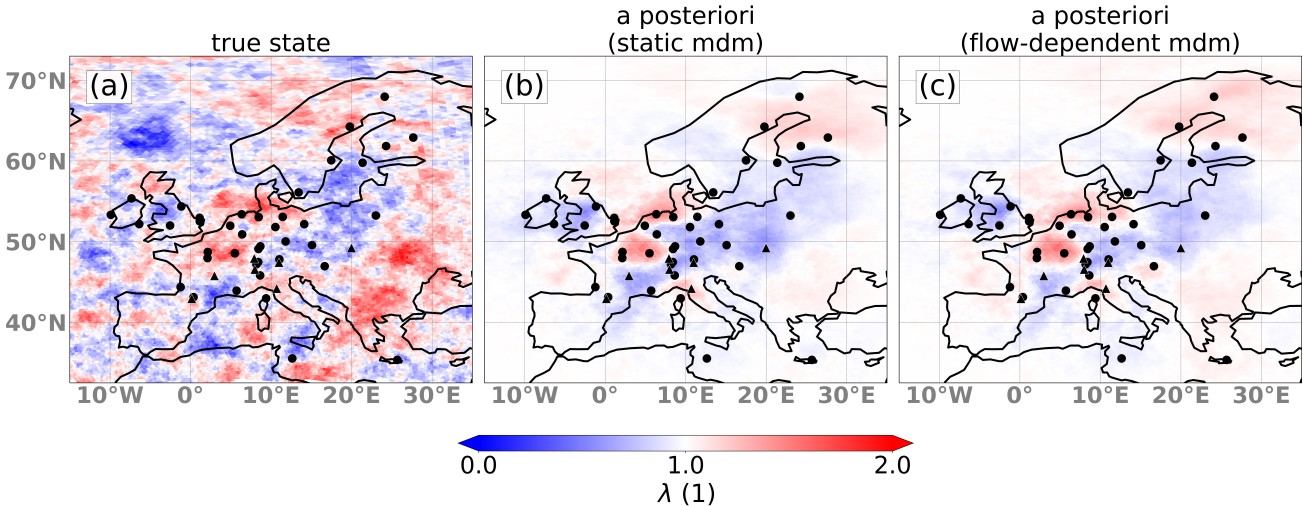

**Figure 8.** True (a) and a posteriori scaling factors for the inversion with the the static mdm (b) and the flow-dependent mdm (c).

variance varies, representing different levels of freedom to adjust the state in the inversion process. The analysis includes both $CH_4$ ("uc01" to "uc16") and $SF_6$ ("us01" to "us16") emission patterns. In the left half of the figure, the a priori variance is larger than the true variance. As a result, the system has too much freedom resulting in strong adjustments of the state. This tendency to overfit leads to poor performance, particularly evident with the static mdm, where the error may become even larger than the a priori error due to over-fitting to biased observations. The right half of the figure shows situations where the a priori uncertainty is too low and the cost of state adjustment is correspondingly high. In contrast to the previous situation with too high uncertainty, the degradation in performance is comparatively slow, and, in some instances, there may even be a slight performance gain, notably with the static mdm. This phenomenon can be attributed to the fact that decreasing the a priori uncertainty minimizes updates in all regions due to higher associated costs in the cost function. As a result, regions initially subject to incorrect updates remain closer to the a priori state. This partially counteracts the performance degradation resulting from reduced updates in regions that initially perform well. This effect becomes apparent as we used (correlated) random perturbation factors for the ground truth, which are normally distributed around 1. In this case, the solution frequently benefits from maintaining proximity to the a priori state. However, in scenarios featuring substantial biases within the a priori on a larger scale, this proximity would likely result in a more rapid decline in performance, as depicted on the right side of the plot.

In the reference setup with a true variance of 0.07, the transport error is relatively large compared to the emission signal. While this makes optimization challenging for the inversion system overall, it presents a greater potential for improvement with the flow-dependent mdm. To evaluate the improvement with the flow-dependent mdm across various magnitudes of the emission signal, we illustrate the relationship between the relative reduction of errors (in percentage) and the true variance ($\sigma^2_{true}$) in Fig. 11. In all these inversions ("fc01" to "fc12" and "fs01" to "fs12"), a perfect assumption is made for the a priori

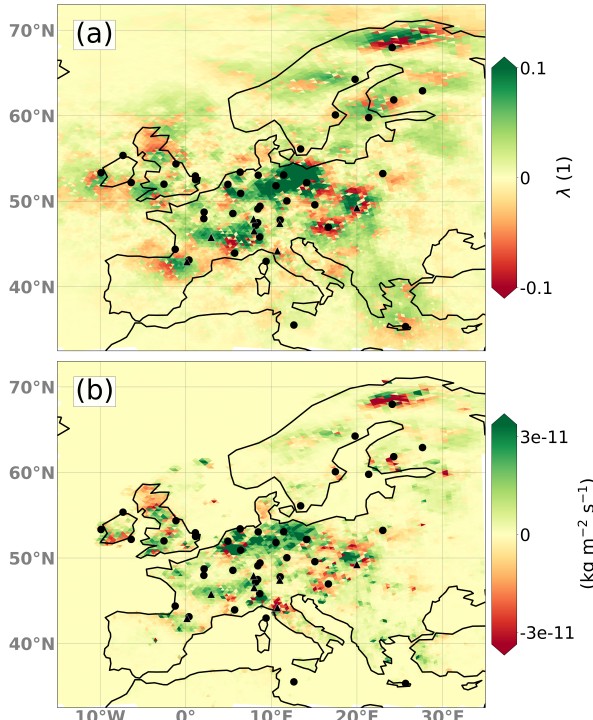

**Figure 9.** Improvement of the error reduction for inversions using the flow-dependent mdm vs the static mdm in terms of scaling factors (a) and emission flux (b). Green color indicates that the inversion with the flow-dependent mdm performs better while red color indicates that the inversion with the static mdm performs better.

variance, resulting in $\sigma^2_{prior}$ being equal to $\sigma^2_{true}$. Moving to the right side of the figure towards larger $\sigma^2_{true}$ values, the ratio of emission signal to transport error increases, making the system a simpler problem to optimize. This is reflected in larger error reductions. However, as the flux signal increases, the distance between the lines representing the static and flow-dependent mdm reduces, i.e. the benefit of applying a flow-dependent mdm becomes smaller. The error reduction of the inversions is generally better for the $SF_6$ emission pattern compared to the $CH_4$ emissions and the flow-dependent mdm has a larger effect. This is to be expected, as the transport error translates into a larger tracer concentration error when the emission pattern is more heterogeneous, as in the case of $SF_6$ emissions.

## 3.3 Hourly observation with correlated errors in an idealized setup

To evaluate the effectiveness of assimilating hourly values versus only daily afternoon mean values (nighttime for mountain sites), we performed 5 different inversions, each using a different R-matrix. All 5 inversions are performed with the $CH_4$ emission field with the true as well as the a priori variance being 0.07. The results are summarized in Fig. 12, which shows the total error reduction across the domain for these inversions (positive y-axis), as well as the reduction in the a posteriori uncertainty

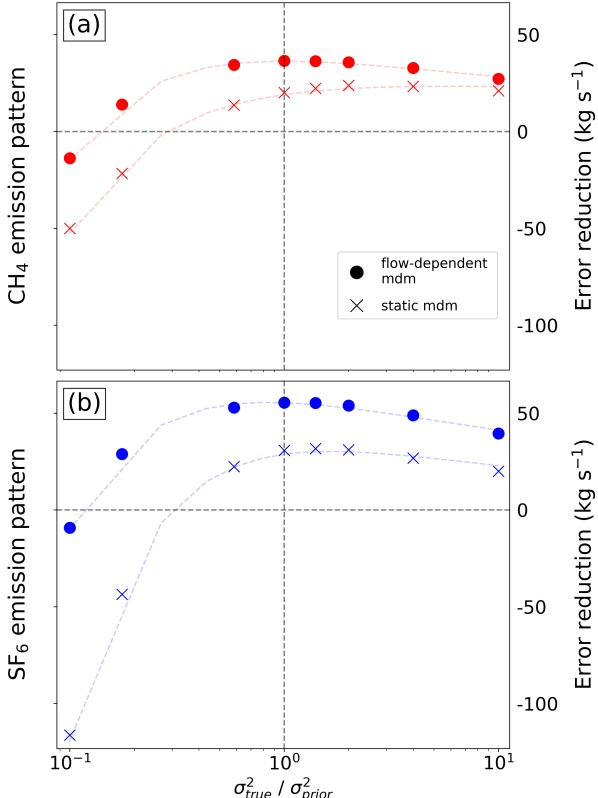

**Figure 10.** Error reduction (in $kg\,s^{-1}$) for inversions using the flow-dependent mdm (circles) and the static mdm (crosses) as a function of the ratio of true variance to a priori variance. Results are shown for inversions with $CH_4$ emissions (a) and $SF_6$ emissions (b).

compared to the a priori uncertainty (negative y-axis). The first two boxes show the error reduction of the two inversions assimilating daily afternoon and night averages, both with constant and flow-dependent mdm ($R_a$ and $R_b$, respectively). These two inversions correspond to the two points on the far left in Fig. 11. The next box in Fig. 12 represents an inversion with hourly instead of daily observations with a diagonal R matrix with constant values for each station, corresponding to the static mdm ($R_c$). The next box represent an inversion with also a diagonal R matrix but with time-varying elements sampled from the meteorological ensemble, which corresponds to the flow-dependent mdm ($R_d$). The last box represents an inversion, where temporal covariances are included in the R matrix as off-diagonal elements based on the sampled correlations between the hourly observations ($R_e$) as described in Sec. 2.7. Compared to the error reductions of 9.0% and 15.6% for the simulations $R_a$ and $R_b$ with daily observations, inversions assimilating hourly observations exhibit an improved error reduction of 15.8% ($R_c$), 17.7% ($R_d$), and 19.9% ($R_e$). Thus, besides the overall improvement, the performance of inversions assimilating hourly observations also increases when a flow-dependent mdm is used, and it further significantly improves if temporal covariances are considered. These results are consistent with those of Ghosh et al. (2021). In their synthetic study using a dense observational

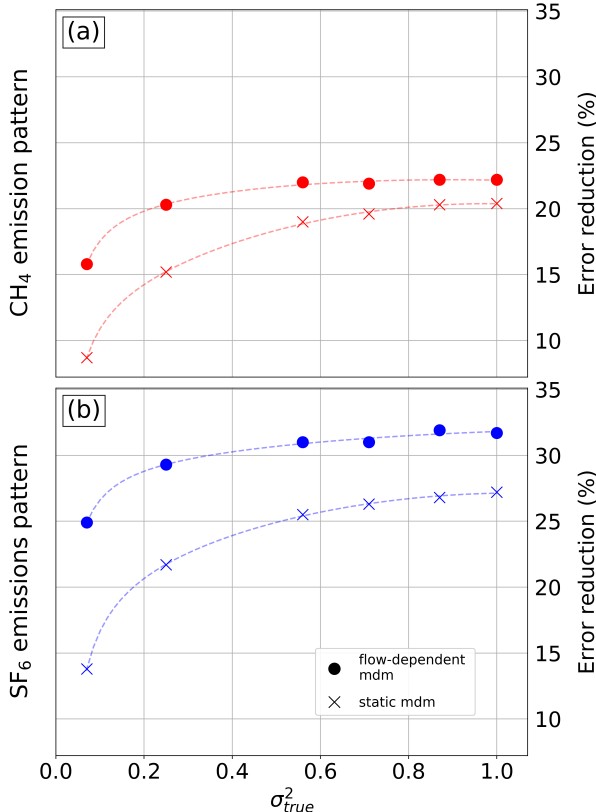

**Figure 11.** Relative error reduction for inversions using the flow-dependent mdm (circles) and the static mdm (crosses) as a function of the true variance. All inversions use the same a priori covariance as the true variance. Results are shown for inversions with $CH_4$ emissions (a) and $SF_6$ emissions (b).

network in an urban area, they observed significant improvements in domain total emission estimates for inversions using a diagonal R-matrix constructed using the ensemble spread (equivalent to $R_c$). They found that non-diagonal R-matrices that account for covariances (such as $R_e$) resulted in better estimation of the spatial emission structure, but this effect diminished when fewer stations were assimilated, which more closely resembles our widely spaced station setup. The negative y-axis in Fig. 12 shows the reduction in uncertainty in the a posteriori P-matrix compared to the a priori uncertainty. Assimilating hourly data results in a significantly larger reduction of the uncertainty in the P-matrix as a greater number of observations is used. When comparing the two inversions that assimilate daily observations, it is evident that the one utilizing the flow-dependent mdm ($R_b$) shows a slightly smaller reduction in uncertainty, despite its much better error reduction compared to the constant mdm ($R_a$). This highlights that uncertainty reduction does not necessarily correlate with inversion performance. Instead, it is a result of assumptions made about error characterizations and the amount of information provided by observations used in the inversion. The same observation also applies when comparing $R_c$, $R_d$, and $R_e$. Here it can be argued that the state of inversions

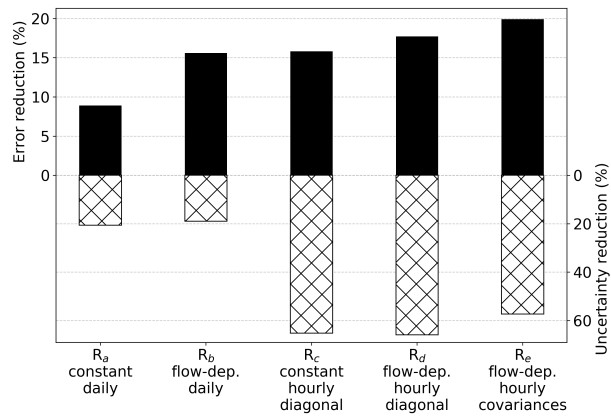

**Figure 12.** Box plot of the domain-total error reduction (positive y-axis) and uncertainty reduction of the a posteriori P-matrix compared to the a priori P-matrix (negative y-axis) of the 5 different inversions ($R_a$-$R_e$).

that assimilate hourly data but lack temporal covariances ($R_c$ and $R_d$) is adjusted too much to the observations, thus suffering from over-fitting.

## 3.4 Effect on real emission estimates

Figure 13 compares the results for the inversions ("real_f" and "real_c") with real data for the month of July 2019. The maps of the increments generally show a very similar pattern, with differences between the two a posteriori emissions shown in Fig. 13c. The significant downward correction over Italy obtained with the static mdm is attenuated with the flow-dependent mdm, as is the case for the Moldavia/Romania region. Similarly, the upward correction over southern England is damped. In contrast, the strong upward correction over the Benelux countries is further enhanced. While the spatial patterns of adjustments differ significantly, domain-total emissions are very similar: The flow-dependent mdm results in total a posteriori emissions of $988 \, \mathrm{kg \, s^{-1}}$, the static mdm in emissions of $981 \, \mathrm{kg \, s^{-1}}$ (with an a priori of $1150 \, \mathrm{kg \, s^{-1}}$ in both cases). In Fig. 13d we present these differences in relation to the a posteriori uncertainty of the inversion using the static mdm. It is evident that there are only few regions where the differences exceed the a posteriori uncertainty, but in many regions this ratio is close to 1. Since the true emissions are unknown in this case, it is impossible to tell which one of the two results is closer to reality. However, based on the results from the synthetic experiments, the results obtained with the flow-dependent mdm are to be preferred. The approach using a static mdm is more likely to assign too much weight to an observation collected during a meteorologically uncertain situation and, conversely, too little weight during a situation when the meteorology is well predicted.

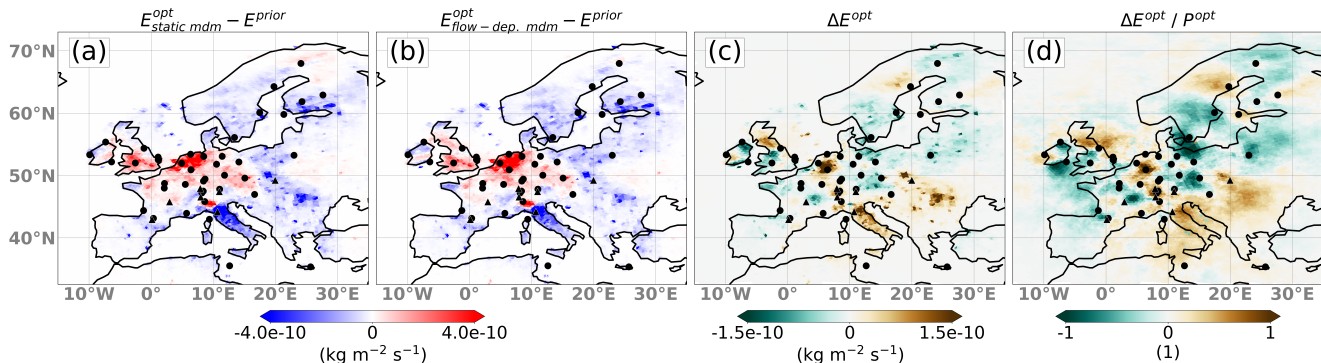

**Figure 13.** A posteriori increments for the inversion with the static mdm (a) and the flow-dependent mdm (b). The difference between the a posteriori emissions for inversions with the flow-dependent mdm and with the static mdm is shown in (c) while the same difference divided by the a posteriori uncertainty is shown in (d).

## 4   Conclusions

385   This paper presents a comprehensive examination of flow-dependent transport uncertainties in GHG inversions. Leveraging meteorological ensemble simulations, we investigate the influence of realistic, temporally varying transport uncertainties on inversion results across various setups and flux strengths and compare it to the more traditional static approximation of the transport uncertainty.

**Error structure characteristics**

390   The spatial structure of transport uncertainty exhibits highly variable patterns, especially when considering the tracer of emitted $CH_4$. This underscores that a static mdm is a poor approximation of the real transport uncertainty. In contrast, the uncertainty in background signal shows a larger-scale, more homogeneous structure. Aside from isolated weather situations, transport uncertainties are typically greatest during the night, reflecting the challenges models face in simulating low nocturnal boundary layer heights. This reaffirms the prevalent practice in current inversion systems of disregarding nocturnal observations or even

395   assimilating only afternoon values at lowland stations. However, it is also apparent that such an ensemble simulation provides the opportunity to filter observations based on the corresponding uncertainty in the model rather than the time of day, and even enabling the assimilation of more observations. Similar to Miller et al. (2015), we couldn't find a clear correlation between transport uncertainties and wind speed that would support the approach of Bergamaschi et al. (2022) of assigning larger uncertainties to observations under low wind conditions. However, with our synthetic setup, it was not possible to test their plausible

400   hypothesis that local sources not represented by the model due to insufficient resolution have the largest influence when wind speeds are low.

**Flow-dependent mdm in inversions**

We compare the inversion results with a flow-dependent mdm and the standard static mdm in an idealized setup with synthetically produced observations. Both inversions show the best performance in central Europe, where emission fluxes and observation density is large. However, inversions with the flow-dependent mdm achieve a larger overall improvement. Depending on the flux signal and emission pattern, relative improvements from 9% (for the largest flux signal of $CH_4$ emissions) to 81% and 82% (for the smallest flux signal of $SF_6$ and $CH_4$ emissions, respectively) are achieved. However, it is crucial to note that while our study achieved large improvements with the new error description, these advancements were observed within an idealized, synthetic setup where the (artificial) transport error is internally consistent with the uncertainties derived from the ensemble spread. Furthermore, we present an analysis where we depart from the assumption of perfect a priori uncertainty in inversions and highlight the importance of making realistic assumptions about a priori uncertainty as well. In particular, overestimating the a priori uncertainty quickly leads to over-fitting to the biased observations. Conversely, being too conservative in the assumptions of a priori uncertainties results in a less pronounced decrease in performance, at least as long as the a priori assumptions do not have large-scale biases.

**Assimilation of hourly observations**

In this study, we also assessed the effectiveness of inversions assimilating hourly versus daily observations. For hourly observations it was necessary to account for temporal correlations in the transport error and hence to include off-diagonal elements in the R matrix. Our analysis showed that these correlations typically exhibit an exponential decay with time, with nighttime observations showing more persistent correlations within the same night. However, we observed a notable exception at lowland stations, where correlations increased again during the following night, peaking at values of 0.5. To incorporate these correlations into our R-matrix, we fit a function to the sampled correlations. This function had to be damped with an exponentially decaying factor to facilitate robust results for the inverse of the otherwise ill-conditioned matrix $\mathbf{HP^bH + R}$. The results demonstrate that hourly data assimilation leads to superior performance compared to daily assimilation of observations. Within the inversions with hourly observations, the performance improved when a flow-dependent mdm was used instead of a constant mdm and it improved even more when in addition temporal covariances were considered. The results also indicate that uncertainty reduction in the P-matrix does not necessarily correlate with inversion performance. It is rather a result of assumptions about error characterizations and the amount of information provided by observations used in the inversion.

**Inversion with real observations and its limitations**

Examining the inversion results with real observations for July 2019, we find that in certain regions, such as Italy and Moldavia/Romania or southern England, the flow-dependent mdm attenuates either the downward or the upward correction, while in other regions, such as the Benelux countries or Switzerland, the upward correction is amplified. However, it is noteworthy that these differences, although often comparable in magnitude to the uncertainties, rarely reach significance relative to the a posteriori uncertainty. We have applied the flow-dependent mdm to real observations assuming that the results are improved compared to the static mdm in a similar way as in the idealized setup. However, while the ensemble spread in the idealized setup accurately captures all transport uncertainties, this is not guaranteed in a setup with real data. Further analysis would be

needed to verify whether the ensemble spread provided by the ECMWF EDA NWP system adequately reflects the differences between the measurements and the simulations. Furthermore, in the idealized setup only random uncertainties are accounted for, whereas in reality there may also be systematic transport errors. Systematic errors could result, for example, from a mis-representation of vertical mixing in stable nocturnal boundary layers, which are known to be particularly difficult to simulate and, at the same time, to have a large impact on near-surface concentrations. While the assimilation of hourly data provided improved results in the idealized experiment, the preliminary result for real measurements revealed unexpectedly large differences from the results obtained with the assimilation of daytime data only. These real-world tests with hourly data raised concerns about the inclusion of nighttime observations in particular, which are very challenging for the model to represent correctly. Accurate inversion depends on the ensemble spread reliably capturing uncertainties and the model being free of significant nighttime biases. Given the need for further investigation of these issues, we have limited our demonstration to the afternoon/night averages to ensure more robust conclusions.

While the assimilation of hourly data provided improved results in the idealized experiment, the result for real measurements was less convincing. Preliminary tests with real hourly data raised concerns about the inclusion of nighttime observations in particular, which are very challenging for the model to represent correctly. Accurate inversion depends on the ensemble spread reliably capturing uncertainties and the model being free of significant nighttime biases. Given the need for further investigation of these issues, we have limited our demonstration to the afternoon/night averages to ensure more robust conclusions.

In conclusion, our findings demonstrate the advantages of integrating temporally varying, flow-dependent atmospheric transport uncertainties in inversions to enhance the accuracy of GHG flux estimations. Incorporating these uncertainties yields more accurate estimates of GHG emissions, with significant improvements across a wide range of setups.

*Code and data availability.* Access to the git repository containing the official CTDAS code can be obtained by contacting the main developers. The code version used for this study is available in a separate branch of the repository. ICON is an open-source modeling framework available with a Digital Object Identifier (DOI): doi:10.35089/WDCC/IconRelease01. Our implementations in the code are available in a separate branch of the icon-kit repository at the dkrz gitlab.

*Author contributions.* MS and DB initiated the project. LC conducted the ensemble simulation experiment with perturbed CH4 at ECMWF and provided the data essential for this study. MS performed and analyzed the simulations and inversion under the supervision of DB. MS, DB and SH interpreted the results. MS wrote the paper with substantial contribution of DB and SH.

*Competing interests.* The authors declare that they have no conflict of interest.

*Acknowledgements.* The Center for Climate Systems Modeling (C2SM) at ETH Zurich is acknowledged for providing technical and scientific support. ICON-ART simulations were performed at the Swiss National Supercomputing Cetre (CSCS) under the grant s1152.

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
