# Peer review of "Flow-dependent observation errors for GHG inversions in an ensemble Kalman smoother"

_EGUsphere, 2024_

## Author Response (AR1)

**Reply to the comment of Referee 1**

We would like to express our gratitude to the referee for the effort in reviewing our manuscript and for the constructive feedback. We have considered all comments and suggestions and address each point in this response letter, with reviewer comments in blue and author responses in black. We believe that the corresponding revisions to our manuscript will enhance its clarity, accuracy, and overall quality.

Steiner et al. investigated the impact of flow-dependent transport uncertainties on the methane inversions and found that in the OSSE setting that this approach of specifying more realistic errors improves the accuracy of the inversion. The model transport error is an important yet often underdiscussed factor in the inverse modeling, hence the method proposed here is useful for improving error specification in regional-and continental-scale inverse modeling. Below are my comments, mostly suggestions for clarification. The paper is publishable once they are addressed.

L6 and L16: L6 : flow-dependent method improves accuracy, L16: improves precise. It appears that most of the discussion focuses on accuracy rather than precision.

This is correct, and we have adjusted the terminology accordingly so that we now use accuracy instead of precision.

L99-102: the description here is confusing. L99: "we optimized only the emissions, but not the background..". L102:"optimization of background levels together with emissions". Some clarification is needed.

In the idealized setup, we did not optimize the background concentrations because the errors in the background (due to the use of different meteorology and different background concentrations) were part of the artificial transport error. Consistent with this, the ensemble spread caused by the different background concentrations contributed to the overall ensemble spread that determined the mdm. However, in a real data application where non-random biases may occur, it is still useful to optimize the background. This is what we did in our real data application.
We have revised the sentences for clarity as follows:

"In contrast to our previous study, we optimized only the emissions in the idealized setup. We did not optimize the background concentrations because the differences in background $CH_4$ concentrations introduced by deviating from ''true'' meteorology and using perturbed background $CH_4^{bg}$ concentrations in the driving data are part of the artificially created transport error and contribute to the ensemble spread that determines the mdm. However, if there were systematic biases in background $CH_4$ in an application with real data, it would still be necessary to optimize background concentrations together with the emissions."

L125-126: (1) In most part of the paper, standard deviation of the ensemble is used (e.g., fig.4). But here "ensemble spread" is used. Are they the same thing? (2) Would this procedure mostly exclude nighttime observations? This information may be useful for readers to assess the difference of this treatment from using daytime only observations.

Yes, in line 125, by "ensemble spread," we meant the standard deviation. We have updated the terminology to "standard deviation" for clarity.

The following figure shows the fraction of excluded observations per hour of the day (from UTC time). In fact, more observations are excluded during the night hours, but also about 15% of daytime observations are excluded. A rough division into day (08 - 20 UTC) and night (all other hours) shows that 1735 day observations and 2514 night observations are excluded. We have added this figure to the manuscript.

[Figure]

L143: What is SPPT? What "observations" is referred to here? Weather observations, or GHG observations? Spell out SST as well.

We have clarified that SPPT stands for "Stochastically Perturbed Parameterization Tendencies" and added this explanation to the manuscript along with the appropriate reference. We have also spelled out SST. In line 143, "observations" referred to weather observations. However, also the GHG observations are perturbed, and we have now included this information in the text.

L170: The equation is not formally written, by mixing the matrix/vector form with the scalar form. Better to write in the matrix/vector form. 1/n (y-Hx)T (HPH+R)^-1 (y-Hx)

[Figure]

Thank you for bringing this to our attention. We have corrected the formula accordingly.

Section 2.6. A lot of discussion regards to flat-terrain sites vs. mountain sites. But no information is provided in Fig. 7 or 12 which sites are classified as flat-terrain or mountainous. It would be good such information can be provided.

We've marked the mountain stations with triangles in Figures 7, 8, and 12 and we note this in Section 2.6 of the manuscript.

L220: Only temporal correlation is considered in this study. However, Fig. 3 indicates that transport model may also cause "flow-dependent" spatial error correlations, by comparing the error patterns in Fig. 3 and the site distribution in Fig. 12. It may make sense to derive this spatial error correlations from the perturbed physics ensemble, and assess the impact of spatial error correlation on the inversion results.

The referee is right that these spatial correlations could also be sampled from the ensemble. However, due to the small number of ensemble members, these correlations are very noisy. One would have to apply a diffusion model, as Lauvaux et al. (2009) did, to extract the significant part of the noisy error. Their diffusion model is based on the method of Pannekoucke and Massart (2008), which estimates the structural properties of the ensemble variability using a diffusion operator based on a local diffusion tensor. While we have considered such an approach, we have decided against implementing it due to the considerable effort involved, especially given our setup covering the large domain of Europe, where spatial distances between stations are often (though not always) large, making temporal correlations much more critical than spatial correlations in our application.

L225-226: For conditioning the R-matrix, what is the reasoning to apply an exponential decay of exp(-deltaT/24h)? Would this defeat the purpose of error correlation fitting performed above?

A detailed rationale for the use of shrinkage methods or smoothing of off-diagonal elements can be found in Ghosh et al. (2021). Spurious correlations arise due to the small ensemble size, resulting in poor conditioning of the R matrix when considering only the sampled covariances in the off-diagonal elements. As described in the manuscript, this leads to instability and produces completely unrealistic values. Therefore, Ghosh et al. (2021) proposed two different approaches. In addition to smoothing by exponential damping of the off-diagonal elements, which we also use, they applied a shrinkage-based regularization of the covariances. Both approaches yielded similarly good results, although they inherently reduce the influence of true correlations to some extent. While correlations with observations in the following hours remain well preserved, correlations

between nighttime observations tend to be significantly attenuated, affecting their reappearance in the following night. Nevertheless, as shown in our synthetic setup, our results still show improvement, allowing us to extract more information from the observations overall. We also tried a longer length scale for the damping factors of 30 hours instead of 24 hours, but this again led to unstable results in the inverse of **HPH+R**.

Fig. 3: the unit of Fig. 3 is kg/kg, which is an uncommon unit for CH4, and is inconsistent with the unit (ppb by volume) used elsewhere in the paper.

We have changed the unit in the figure correspondingly.

Fig. 5: This figure shows interesting information. But I am confused why in panel c and d, the STD(CH4_emis) is non-zero for the perturbed IC/BC only case. If I understand correctly, CH4_emis tracer is not affected by IC/BC but changes with wind fields in the modeling domain. In the perturbed IC/BC only case, the wind fields within the modeling domain are identical across the ensemble member, and, therefore, should result in the same CH4_emis tracer.

"Perturbed IC/BC only" means that the ensemble experienced only perturbed initial and boundary conditions, without additional perturbation by perturbed model physics. Perturbed initial and boundary conditions mean that the driving meteorological data (and the $CH_4$ field used to drive the background) are perturbed, resulting in each ensemble member having a different meteorology. This leads to differences in the emitted $CH_4$ tracer. We have revised the text to clarify that perturbed IC/BC refers to the way the ensemble is generated, not just the $CH_4$ concentration field. Specifically, we have changed the following sentence:

"Each figure presents two scenarios: one with only perturbed initial and boundary conditions (gray), and the other one with both, perturbed model physics and perturbed initial and boundary conditions (IC/BC, red)."

to

"Each figure presents two scenarios: one where the ensemble is generated with perturbed initial and boundary conditions only (IC/BC, gray), and one where in addition to IC/BC also the model physics is perturbed (IC/BC + STTP, red).".

L317: and -> with

Corrected

[Figure]

Section 3.4: It appears that the real-data inversion uses only daytime average data. Is it possible to use hourly data filtered by flow-dependent errors, as shown in Section 3.3?

This is technically possible. However, while the assimilation of hourly data yields better results in OSSE, it is unclear whether this holds true for real measurements. This would require that the ensemble spread accurately reflects the real uncertainties, and that the ensemble simulation does not have substantial biases at night. The latter is probably the more critical issue. Therefore, we refrain from including these results in the publication and limit our demonstration to the idealized setup.

Section 4: This long paragraph is difficult to read. I'd suggest to organize the section into separate paragraphs.

We have divided the Conclusions into individual paragraphs, each with its own title.

L395-403: This interesting discussion should belong to "Results and Discussion" instead of Conclusion.

We have moved most of this discussions to "Results and Discussion", and briefly mentioned it again in the Conclusions.

[Figure]

**Reply to the comment of Referee 2**

We would like to thank the referee for taking the time to read our manuscript and for the thoughtful feedback. The input was very valuable to enhance the quality and rigor of our work. We have taken all suggestions along with the comments of Referee 1 into account and present our responses below, with reviewer comments in blue and author responses in black.

**General comments**

The paper describes experiments with a new implementation of the observation-representation-error in an Ensemble Kalman smoother system that is used for emission inversion of methane. The new method uses ensemble simulations to quantify the transport uncertainty at observation sites from the ensemble spread. The ensemble perturbations are driven by a meteorological ensemble and model changes, and therefore time- and flow-depended.

The authors show that the new dynamic formulation of the observation-representation-error (or as called in the paper, 'mdm' or 'model-data-mismatch') is beneficial over an implementation using a static error. It allows for example the use of CH4 observations for every hour of the day rather than afternoon or nighttime averages only as is currently common practice, with improved result. Although running an *a priori* ensemble is not cheap, it seems worthwhile the effort. This is an important conclusions, and the paper is therefore a useful contribution to the field of greenhouse-gas inverse modelling. Could the authors give an indication of the costs (computational, storage, work?) of the proposed method? Compared to regular inversions, are the costs minor or substantial? Could external data suppliers be of use here and make dedicated data available for the purpose of this method?

The computational cost, measured in node-hours, for the a priori ensemble simulation with 10 members (without the flux ensemble with 192 tracers) was about 1.7 times higher than the cost of regular inversions. This means that the total computational cost (prior ensemble + inversion) was 2.7 times the cost of a regular inversion, which is a considerable but not prohibitive increase. We recognize that this important information should be communicated in the manuscript, and have therefore added it to the Methods section (subsection "ICON-ART ensemble simulations").

In terms of storage, the output of the a priori ensemble simulation required about half as much space as the output of the regular inversion. Thus, the storage requirement was 1.5 times the size of a regular inversion.

It is important to emphasize that we did not optimize the a priori ensemble simulation for computational time and memory. We wrote out significantly more variables than were necessary to determine the mdm alone, so as not to limit our analysis capabilities for this study. It is likely that the additional cost of the a priori ensemble simulation could be reduced.

[Figure]

Materials Science and Technology

The paper summarizes results from many experiments, and therefore often requires re-reading of previous sections to capture what exactly has been done. The understanding would increase if the authors could include a table with all experiments and how their configuration is different, for example using the keywords R_a to R_e as used in Figure 11. In addition, a figure illustrating the data flow would be useful, as there are many entities used that could be confused with each other. There is for example the Kalman Smoother ensemble but also a meteorological ensemble; there are synthetic observations used but also real data; boundary conditions originate from ERA5, ICON, and CAMS; tracers could be either CH4 but also SF6; etc. A better overview of the data stream would help to distinguish the various elements from each other, and also help interested readers to implement a similar system themselves.

We have summarized all the inversions presented in this study in two tables. In the text, we now refer to the inversion IDs used in these tables. In addition to the ID, the tables list the type of mdm (flux-dependent or constant), the emissions used, the a priori and true variance, the scaling factor alpha used to achieve an innovation chi-squared value of 1, and the error reductions. For inversions with hourly and/or real observations, this is noted as a comment. However, we decided not to include an additional graphic to illustrate the entire data flow. We couldn't come up with a clear and concise schema that accurately depicted all the entities. Nevertheless, we believe that the restructuring and revisions made to the text in response to the reviewers' comments and the inclusion of the table have resulted in a clearer and more understandable manuscript.

**Specific comment**

Line 29: "Errors in the second term, commonly referred to as model-data mismatch error (mdm),".

A more common description is (I think) that **R** is the "observation representation error".

The matrix **R** is indeed often described as the observation representation error. Other papers refer to it as the model-data mismatch or simply as the observation error. We prefer the term model-data-mismatch here because a large part of the error is neither due to observation errors nor a representation problem, but rather due to model errors. To avoid any confusion, we now introduce the term as follows: "Errors in the second term, commonly denoted as **R** and referred to as observation representation error or model-data mismatch error (mdm), include all processes…".

Line 68-69: The term "OSSE" is more often used for to assimilation experiments that should quantify the impact of observation instruments or networks that do not exist yet. Not sure if the experiments with synthetic data that are described here could be described as such, but this is of course only a detail.

It is true that the term "OSSE" is often used to describe assimilation experiments designed to quantify the impact of proposed observing instruments or networks. Although our goal was different, the approach was the same, i.e. defining a true state (emissions) generating synthetic pseudo-observations, and performing inversions with these observations. It has become quite common in the community to use the term "OSSE" for such experiments.

**Lines 87-88: What perturbed the CH4 boundary conditions? This seems described later on in section 2.3. Bud did the global ensemble use the same emissions and only different meteorology?**

The CH4 concentrations in the global ensemble are perturbed not only by the differences in meteorology, but also by perturbed emissions, as detailed in Sect. 2.3. To clarify this at the point you mention in the text, we have added the following sentence:

"... as well as by perturbed $CH_4$ boundary conditions from the same global ensemble simulation that provided the meteorological boundary conditions. *In this global ensemble, the $CH_4$ concentrations are perturbed as a result of perturbed meteorology and perturbed emissions.*"

**General: The order in which various elements of the system are introduced is more top-down than bottom-up now. Maybe this could be changed? For example: 1. model description (first paragraph under section 2); 2. meteorological ensemble (subsection 2.3); 3. ensemble simulations (subsection 2.1).**

We have adjusted the structure as suggested. We have moved sections 2.3 (ICON-ART ensemble simulations) and 2.4 (CTDAS inversion setup) before the description of the two experiments and revised the text accordingly. We believe this restructuring makes the text more understandable.

**Lines 103-104: "To ensure a fair comparison, inversions were set up so that the innovation chi-squared value in each inversion was 1". What does this mean, that the parameters that define the *a priori* model uncertainty are tuned to ensure Chi^2=1? Which parameters are these? It seems explained later in Eq.(1).**

Later in the text, we describe that we achieve a chi-squared value of 1 by scaling the mdm and explain the reasoning behind this. To make this clearer already at this point, we have revised the sentence as follows:

"" -> "To ensure a fair comparison, the mdm was scaled in each inversion so that the innovation chi-squared..."

**In Eq.(1) the ratio is between scalars, but text mentions that R is a "covariance". Shouldn't it be actually a vector-matrix-vector product:**

(yo-H(xb))^T (HPbH+R)^{-1} (yo-H(xb))

Yes, we corrected that.

Figure 6: The caption could include part of the description of lines 265-266 to know what the different lines are.

We have added the following sentence to the figure caption:

"Mean temporal correlations at Cabauw (a and b) and Monte Cimone (c and d) for observations at 00 to 08 UTC (a and c) and 12 to 20 UTC (b and d) with observations in the next 36 hours. *The error correlations over time (x-axis) for each observation is indicated by one line.*"

**Technical corrections**

The manuscript is well written and easy to read, so only a few errata found.

Line 105 and more: usually "*a priori*" and "*a posteriori*" are written in *Italic* font.

The ACP English Guidelines and House Standards explicitly state that common Latin phrases such as "a priori" and "a posteriori" should not be italicized. Therefore, we have kept the current formatting in the manuscript.

Line 129: .. observation*s*, ..

Corrected.

---

## Editor Decision (ED1)

**Editor comments on egusphere-2024-1426**

**Comments to your reply to the referee comments:**

1. I think the term ensemble spread should be still used since this is a commonly used term in the meteorological/assimilation community. Instead you should explain somewhere that it corresponds to the standard deviation.
2. Section 3.4: If you refrain from adding these results in the manuscript, doesn't this mean that you have the results already? In your first sentence you state that you do not know the outcome.

**General comments on the manuscript:**

1. Consider restructuring Section 2. Ten pages are needed to reach the result section. Make subsections for the real and idealized experiments, where the respective data sets are described.
2. Discuss better the limitations of the experiment with real data. How can these be optimized? Wouldn't this be the aim? Which regions have been considered? Europe? You should clearly state in the text which regions are considered.

**Specific comments on the revised manuscript:**

L89: Why perturbed? Do you mean unperturbed? Isn't the perturbation of the background coming from the emissions?

L91,92: Why does $CH_4^{tot}=CH_4^{bg}+CH_4^{emis}$ represent transport uncertainty?

L94 and L97: "10 ensemble members" and "10 members" written -> repetition?

L105: GOSAT, IASI and TROPOMI -> abbreviations of the instrument names should be introduced.

L106: the "driving data" has not been introduced. Which data is this? $CH_4^{bg}$, $CH_4^{emis}$ or something different?

L139: Here the term "spread" is still used. This contradicts to your reply to the referee comments.

L161-163: Introduce abbreviations used.

L178: real -> What data is included in which experiment? Where is the nighttime data included? Didn't you in your answer state that only daytime data was used for the real experiments?

For the experiment with real observation pseudo observations are used, isn't this a contradiction?

L211ff: Here you use the term standard deviation.

L267: Here "spread" used.

Figure 6 caption: Figure label "std", but caption "spread".

L369: Add which figure.

**Technical corrections:**

L88: Space between full stop and "In the global" missing.

L165: according -> according to

L220: a similar principles -> a similar principle

L291: supports -> support ? Please check the sentence.

L395: add comma to the sentence?

---

## Author Response (AR2)

**Dear Dr. Khosrawi,**

Thank you very much for your positive response and the opportunity to address the minor revisions you suggested. We have carefully considered each of the points and have made the necessary revisions to the manuscript. Below, we provide a detailed response to each comment, outlining the changes made in the revised manuscript.

**Comments to the replies to the referee comments:**

1. I think the term ensemble spread should be still used since this is a commonly used term in the meteorological/assimilation community. Instead you should explain somewhere that it corresponds to the standard deviation.

   - The term "ensemble spread" is used for the first time on L139, where we now added the explanation that it corresponds to the standard deviation:
   "The ensemble spread (corresponding to the standard deviation) of the $CH_4^{tot}$ tracers was sampled at..."

2. Section 3.4: If you refrain from adding these results in the manuscript, doesn't this mean that you have the results already? In your first sentence you state that you do not know the outcome.

   - To clarify, we have indeed carried out preliminary tests with the assimilation of hourly real observations but the results were not convincing as they showed unrealistically large increments and partly very large differences from the assimilation of afternoon data. Although in the idealized setup the assimilation of hourly data gives better results, it remains uncertain whether this is true for real measurements, especially for observations collected in stable nighttime boundary layers, which are very difficult to model. This uncertainty is due to two open questions: (i) whether the ensemble spread reliably reflects the real uncertainties, and (ii) whether the reference simulation is free of significant nighttime biases, the latter being probably the more critical issue.

     A much more detailed investigation of the nighttime model performance would be needed to address these questions, and we felt that this would be outside the scope of the present paper. Including these results without such a detailed analysis could lead to misleading conclusions. Therefore, we decided to limit the scope to the more robust results from the idealized setup.
     We have added a short explanation in the discussion section as to why the hourly data assimilation results are not included, and clarify the limitations or uncertainties associated with the assimilation of real hourly measurements:

"While the assimilation of hourly data provided improved results in the idealized experiment, the preliminary result for real measurements revealed unexpectedly large differences from the results obtained with the assimilation of daytime data only. These real-world tests with hourly data raised concerns about the inclusion of nighttime observations in particular, which are very challenging for the model to represent correctly. Accurate inversion depends on the ensemble spread reliably capturing uncertainties and the model being free of significant nighttime biases. Given the need for further investigation of these issues, we have limited our demonstration to the afternoon/night averages to ensure more robust conclusions."

**General comments on the manuscript:**

1. Consider restructuring Section 2. Ten pages are needed to reach the result section. Make subsections for the real and idealized experiments, where the respective data sets are described.

   - We appreciate your suggestion to restructure Section 2, and we understand the intent to make the content more accessible and structured for readers.
     Given the nature of this manuscript, which extensively tests a new methodology, a detailed description of the experimental setup is essential. The length of the methods section reflects the need for precision in describing the methodological details. As the results section spans 10 pages, we believe that the ratio of methods to results is appropriate for a paper of this type.
     In the methods section, we have already included subsections that distinguish between the idealized and real experiments, as well as subsections on pseudo and real observations. We believe that further subdivisions would not improve readability.
     Thus, after careful consideration, we believe that the current structure already provides a clear and coherent organization of both the real and idealized experiments.

2. Discuss better the limitations of the experiment with real data. How can these be optimized? Wouldn't this be the aim? Which regions have been considered? Europe? You should clearly state in the text which regions are considered.

   - The setup of the real data experiment is very similar to the idealized experiment described in subsection 2.4. This includes the domain over Europe. The ICON grid is described in Section 2 and is valid for all inversions. However, to make this clearer, we have added the following sentence to Section 2.4:

     "We used the same setup for the ICON simulations with a grid over Europe and the same state vector in the inversions.

[Figure]

We further extended the discussion of the real data application with the following discussion about its limitations:

"We have applied the flow-dependent mdm to real observations assuming that the results are improved compared to the static mdm in a similar way as in the idealized setup. However, while the ensemble spread in the idealized setup accurately captures all transport uncertainties, this is not guaranteed in a setup with real data. Further analysis would be needed to verify whether the ensemble spread provided by the ECMWF EDA NWP system adequately reflects the differences between the measurements and the simulations. Furthermore, in the idealized setup only random uncertainties are accounted for, whereas in reality there may also be systematic transport errors. Systematic errors could result, for example, from a misrepresentation of vertical mixing in stable nocturnal boundary layers, which are known to be particularly difficult to simulate and, at the same time, to have a large impact on near-surface concentrations."

**Specific comments on the revised manuscript:**

L89: Why perturbed? Do you mean unperturbed? Isn't the perturbation of the background coming from the emissions?

- We do mean "perturbed" here. The global ensemble we use to drive our simulations contains perturbed meteorology as well as perturbed $CH_4$ concentrations. With these perturbed $CH_4$ concentrations we drive our background tracer at the boundaries. In addition, the meteorology is different for each ensemble member, which also contributes to the ensemble spread in $CH_4^{BG}$. However, no emissions are added to the $CH_4^{BG}$ tracer; this only affects $CH_4^{emis}$.
  In the global ensemble, on the other hand, the perturbed $CH_4$ concentrations result from perturbed emissions, perturbed meteorology and perturbed observations.
  We have removed the sentence explaining how the $CH_4$ concentrations are perturbed in the global ensemble. This is now explained in detail in the following subsection. We think that this makes the text clearer, since we now only refer to the ensemble in our simulations:

"Unlike our previous study with a single forward simulation, we created a meteorological ensemble (see Sect. 2.1) with 10 ensemble members driven by perturbed meteorological boundary conditions and model physics, as well as by perturbed $CH_4$ boundary conditions from the same global ensemble simulation that provided the meteorological boundary conditions.  Each ensemble member contains two $CH_4$ tracers: a background tracer ($CH_4^{BG}$) representing the perturbed $CH_4$ boundary conditions, and an emission tracer ($CH_4^{emis}$) representing the additional $CH_4$ emitted within our European model domain."

**L91,92: Why does CH4tot=CH4bg+CH4emis represent transport uncertainty?**

- With this sentence, we express that the ensemble spread in our simulated $CH_4$ concentrations (and in the end we are interested in the total concentration of $CH_4^{BG}$ + $CH_4^{emis}$) reflects the transport uncertainty and does not reflect uncertainties in emissions due to disturbed emissions. Although, $CH_4^{emis}$ is the result of emissions in the domain, its ensemble spread is only driven by changes in the transport, whereas the emissions are identical for all ensemble members.

**L94 and L97: "10 ensemble members" and "10 members" written -> repetition?**

- In one case, the "10 members" refers to the number of members used in the ICON-ART simulations, and in the other case, it refers to the number of members used in the global ensemble. These do not have to be the same; one could also use a selection of the available members for a simulation. Here, however, we use all 10 available members, which is why the numbers are the same.

**L105: GOSAT, IASI and TROPOMI -> abbreviations of the instrument names should be introduced.**

- We now write out the full names of the instruments and provide the abbreviation in parentheses:

  ""
  "Greenhouse Gases Observing Satellite (GOSAT), Infrared Atmospheric Sounding Interferometer (IASI), and TROPOspheric Monitoring Instrument (TROPOMI) retrievals of $CH_4$ were assimilated."

**L106: the "driving data" has not been introduced. Which data is this? $CH_4^{bg}$, $CH_4^{emis}$ or something different?**

- By "driving data" we mean the meteorological variables and the $CH_4$ concentrations for the boundary conditions of $CH_4^{BG}$, which we describe in Sect. 2. We add this in brackets on line 106 and start the sentence with "In the ICON ensemble simulations" to make this clearer:

  ""

  "In the ICON ensemble simulations, in addition to the perturbed driving data (meteorological variables + $CH_4$ concentrations to drive the $CH_4^{BG}$ tracer), we also applied perturbations to model physics tuning parameters as implemented in ICON for the ensemble data assimilation scheme of the German weather service."

**L139: Here the term "spread" is still used. This contradicts to your reply to the referee comments.**

- We had overlooked "ensemble spread" at this point, when revising the text. However, in line with your first comment, we will leave it at "ensemble spread" and instead add that the ensemble spread corresponds to the standard deviation:

  "$_4$"

  "The ensemble spread (corresponding to the standard deviation) of the $CH_4^{tot}$ tracers was sampled..."

**L161-163: Introduce abbreviations used.**

We introduce the abbreviations JSBACH-HIMMELI, GCP and GFED:

"~~...peatlands and mineral soils from JSBACH-HIMMELI (Raivonen et al., 2017; Reick et al., 2013) (version 2), inland water (provided by Université Libre de Bruxelles to the GCP-CH4 data set; Saunois et al., 2020), ter-mites (Saunois et al., 2020), ocean (Weber et al., 2019), and biofuels and biomass burning (GFED 4.1s; van der Werf et al., 2017) as well as geological emissions (Etiope165 et al., 2019) (scaled to a global total of 15 Tg).~~"

"... peatlands and mineral soils from JSBACH-HIMMELI (Jena Scheme for Biosphere-Atmosphere Coupling in Hamburg coupled to HelsinkI Model for Methane build-up and emission for peatlands; Raivonen et al., 2017; Reick et al., 2013) (version 2), inland water provided by Universite Libre de Bruxelles to the Global Carbon Project (GCP) $CH_4$ data set; (Saunois et al., 2020), termites (Saunois et al., 2020), ocean (Weber et al., 2019), and biofuels and biomass burning from the Global Fire Emission Database 4.1s (GFED; van der Werf et al., 2017) as well as geological emissions (Etiope et al., 2019) (scaled to a global total of 15 Tg)."

**L178: real -> What data is included in which experiment? Where is the nighttime data included? Didn't you in your answer state that only daytime data was used for the real experiments?**
**For the experiment with real observation pseudo observations are used, isn't this a contradiction?**

- In the real data application, we follow the standard setup where we assimilate the afternoon means (or nighttime means for mountain station). This is expressed by the sentence in L178. To make this clear, we have added in parentheses (e.g., as in "fc01") to indicate the corresponding inversion in the synthetic setup, which also assimilates afternoon and night means. The "nighttime" part refers only to the assimilation at mountain stations, as described in Sect. 2.3.
  We do not use pseudo-observations anywhere in the application with real data and it is not clear to us where we would have written this in the manuscript.

L211ff: Here you use the term standard deviation.
L267: Here "spread" used.

- In line with your first comment, we now use the term "ensemble spread" throughout the manuscript, except for the figure captions, which use "std" in their labels. Since we explain at the beginning that by "ensemble spread" we mean the standard deviation, we see no problem in using both terms in the manuscript as synonyms.

Figure 6 caption: Figure label "std", but caption "spread".

- The label and caption are brought into agreement by using "standard deviation" also in the caption.

L369: Add which figure.

- Done, also for L373.

**Technical corrections:**

L88: Space between full stop and "In the global" missing.
L165: according -> according to
L220: a similar principles -> a similar principle
L291: supports -> support ? Please check the sentence.
L395: add comma to the sentence?

- We addressed all 5 technical corrections. The sentence in L88 has already changed due to an adjustment to a previous comment of yours.